# Isothiocyanate-Functionalized Mesoporous Silica Nanoparticles as Building Blocks for the Design of Nanovehicles with Optimized Drug Release Profile

**DOI:** 10.3390/nano9091219

**Published:** 2019-08-29

**Authors:** Gabriel Martínez-Edo, Maria C. Llinàs, Salvador Borrós, David Sánchez-García

**Affiliations:** 1Grup d’Enginyeria de Materials (GEMAT), Institut Químic de Sarrià, Universitat Ramon Llull, Via Augusta, 390, 08017 Barcelona, Spain; 2Centro de Investigación Biomédica en Red en Bioingeniería, Biomateriales y Nanomedicina (CIBER-BBN), 50018 Zaragoza, Spain

**Keywords:** mesoporous silica nanoparticles, regioselective functionalization, drug delivery, isothiocyanate

## Abstract

A straightforward methodology for the synthesis of isothiocyanate-functionalized mesoporous silica nanoparticles (MSNs) by exposure of aminated MSNs to 1,1′-thiocarbonyldi-2(1*H*)-pyridone is reported. These nanoparticles are chemically stable, water tolerant, and readily react with primary amines without the formation of any by-product. This feature allows the easy modification of the surface of the nanoparticles for tuning their physical properties and the introduction of gatekeepers on the pore outlets. As a proof-of-concept, amino-isothiocyanate-functionalized MSNs have been used for the design of a nanocontainer able to release the drug Ataluren. The release profile of the drug can be easily fine-tuned with the careful choice of the capping amine.

## 1. Introduction

Mesoporous silica nanoparticles (MSNs) are currently of great interest in biomedicine owing to their potential as building blocks for the design of multifunctional systems for drug delivery [1,2,3,4,5], and bioimaging [6,7]. To a great extent, the versatility of MSNs is due to their singular geometry, which is characterized by the presence of two distinct domains: The internal mesoporous area and an external surface. While the internal mesopores of the MSNs provide a large volume area for the loading of drugs, the outer surface allows their functionalization with valves [8], ligands [9], or probes [10].

In order to attain the level of complexity needed for the design of “smart” nanocarriers, it is necessary to develop efficient methods for the selective functionalization [11,12] of the two surfaces of the MSNs. The coating of the porous surface with the proper functional group (e.g., amines [13,14,15,16], hydrazones [17]) is crucial for the optimization of the drug loading. Regarding the external surface, this domain is suitable to install stimulus-responsive systems for drug release [18] or attach functionality for cell targeting [19]. From a chemical perspective, the preparation of such complex systems requires clean (free of by-products) and optimized (general) methodologies [20,21]. Hence, the ideal building block for the preparation of MSNs-based carriers should be (i) orthogonally bifunctionalized, (ii) easy to prepare and purify, and (iii) chemically stable and storable.

A reliable methodology to prepare such bifunctional MSNs in a regioselective fashion consists in the derivatization of the outer surface of mono-functionalized MSNs with the pores blocked by the surfactant [22,23,24]. In this context, aminated MSNs emerge as an ideal starting material given the rich chemistry of the amino moiety, which can be easily derivatized by nucleophilic or acyl substitution [25,26]. Owing to their versatility, aminated MSNs have been studied in great detail, and the available synthetic protocols based on the co-condensation methodology allow the fine-tuning of their morphology [27,28] and the localization of the amino groups [29]. Thus, the synthetic problem is reduced to the choice of a proper reactive group for the outer surface of the MSNs [30,31]. Typically, MSNs are functionalized using long and tedious procedures requiring the treatment of the particles with a functionalized trialkoxysilane (e.g., 3-aminopropyltrimethoxysilane (APTMS), 3-mercaptopropyltrimethoxysilane (MPTMS)) in refluxing toluene. Afterward, this material can be properly derivatized with the aid of amidation protocols [32,33], the thiol–yne coupling [34], or the Cu(I)-catalyzed azide–alkyne cycloaddition (CuAAC) [35].

Herein, we report the development of ready-to-use isothiocyanate MSNs, which can be prepared by simple exposure of aminated MSN to a “thiocarbonyl transfer reagent”. The isothiocyanate moiety readily reacts with primary amines to yield thioureas and is compatible with aqueous media. The coupling between isothiocyanates and amines does not require any activating agent or transition metal catalyst [36], which must be carefully removed from the nanoparticles prior to any biological test [37]. In terms of reactivity, the isothiocyanate group is comparable to the isocyanate [38,39], however, the former is not prone to hydrolysis [40]. Hence, it is stable in aqueous media and can be stored [41]. Interestingly, although isothiocyanates are well known as reactive groups in bioconjugation chemistry [42], to the best of our knowledge, isothiocyanate-functionalized nanoparticles have been scarcely described in the literature [43]. In this paper, the efficiency of this new procedure will be assessed and compared with the CuAAC protocol. Furthermore, this methodology will be applied to the synthesis of amino-isothiocyanate MSNs. These nanoparticles will be used for the encapsulation of a model drug (Ataluren). The preparation of such a nanovehicle and the fine tuning of the release of the drug will be discussed.

## 2. Materials and Methods

### 2.1. Materials

Cetyltrimethylammonium bromide (CTAB), tetraethylorthosilicate (TEOS), 3-aminopropyl triethoxysilane (APTES), dry toluene, absolute ethanol, methanol, acetonitrile (ACN), anhydrous dichloromethane (DCM), ethyl acetate (EtOAc), and tetraethylene glycol monomethyl ether were purchased from ACROS; ammonium hydroxide (NH_4_OH) from Fluka (St. Louis, Missouri, USA); 1,1′-thiocarbonyldi-2(1*H*)-pyridone, (2-aminoethyl)trimethylammonium chloride hydrochloride, hydrochloric acid (HCl), trifluoroacetic acid (TFA), sodium bicarbonate (NaHCO_3_), magnesium sulfate (MgSO_4_), copper iodide (CuI), *N*,*N*-diisopropylethylamine (DIPEA), sodium diethyldithiocarbamate, ammonium nitrate (NH_4_NO_3_), potassium carbonate (K_2_CO_3_), cystamine dihydrochloride, Ataluren, citric acid, sodium phosphate (Na_2_HPO_4_·12H_2_O), 3-bromopropionic acid, 2,5,8,11,14,17,20,23,26,29,32,35,38,41,44,47-hexadecaoxanonatetracontan-49-ol, fluorescein-5-isothiocyanate (FITC), and 1,8-bromonaphthalimide from Sigma Aldrich (St. Louis, MO, USA). All the chemicals were used as received without further purification.

### 2.2. Synthesis of Amino MSNs with CTAB (MSN–NH_2_ (CTAB))

MSN–NH_2_ (CTAB) were prepared according to the procedure reported in literature [44]. The synthetic procedure was as follows: firstly, 0.2 g of CTAB were dissolved in 100 mL of 0.2 M NH_4_OH at 60 °C, and 1.6 mL of 0.2 M dilute TEOS (in absolute ethanol) was added with vigorous stirring. After the solution was stirred for 5 h, 1.6 mL of 12% (v/v) APTES (in absolute ethanol) and 1.6 mL of 1.0 M TEOS (in absolute ethanol) were added, followed by vigorous stirring for another 1 h. The solution was then aged at 60 °C for 24 h. Solid samples were collected by centrifugation at 13,000 rpm for 13 min, and then MSNs were washed and dispersed only with deionized water in order to avoid CTAB elimination.

### 2.3. Preparation of Amino MSNs without CTAB (MSN–(NH_2_))

Surfactant templates were removed by extraction in acidic methanol (0.17 g of concentrated HCl in 9 mL of absolute methanol) for 24 h. Complete removal of the surfactant was checked by FTIR by inspection of the disappearance of bands at 2900 and 2800 cm^−1^.

The total quantity of amino groups was estimated as follows. To a solution of 10 mg of MSN–(NH_2_) (CTAB) in 10 mL of dry toluene, 10 mg of FTIC were added. This solution was stirred for 24 h at 23 °C. Solid samples were collected by centrifugation at 13,000 rpm for 13 min and the supernatant was measured at 495 nm to determine the quantity of unreacted FTIC. The difference with the total amount of FITC added gave the amino groups present on the external surface of MSN–(NH_2_) (CTAB). The same procedure was repeated with MSN–(NH_2_), without the surfactant, to determine the total amount of amino groups present on the inner and outer surface. Therefore, in total, there were approximately 8 × 10^−7^ NH_2_ mol/mg MSN, from which 6 × 10^−7^ NH_2_ mol/mg MSN was present on the inner surface of the MSN.

### 2.4. Synthesis of Isothiocyanate MSNs (MSN–(NCS))

Nanoparticles without CTAB (MSN–(NH_2_)) were treated with 30 mL of dry toluene at 50 °C for 24 h. Forty milligrams of the resulting MSN–(NH_2_) were suspended in 35 mL of dry toluene and 95 mg of thiocarbonyldi-2(1*H*)-pyridone (**1**) (0.409 mmol, 12 eq.) were added. The suspension was stirred for 24 h at room temperature. Solid samples were collected by centrifugation at 13,000 rpm for 13 min and then MSNs were washed and dispersed with 10 mL of dry toluene and 10 mL of absolute EtOH. This procedure was repeated six times and, finally, the solvent was evaporated under reduced pressure and MSN–(NCS) were stored dry.

### 2.5. Functionalization of MSN–(NCS) with 4-(n-Butylamino)-N-(2-Aminoethyl)-1,8-Naphthalimide (MSN–(UNaph))

Twenty milligrams of MSN–(NCS) were suspended in 15 mL of absolute EtOH. Then 6.6 mg of 4-*n*-butylamino-*N*-(2-aminoethyl)-1,8-naphthalimide (**2**) (0.021 mmol, 2 eq.) were added [45]. The mixture was stirred in the dark for 48 h at room temperature. Solid samples were collected by centrifugation at 13000 rpm for 13 min and washed and dispersed in absolute EtOH until the disappearance of yellow color (five times, confirmed by tlc of the supernatant).

### 2.6. Synthesis of Azido MSNs (MSN–(N_3_))

Forty milligrams of the resulting MSN–(NH_2_) were suspended in 20 mL of absolute EtOH. Afterward, 86 mg (0.405 mmol, 12 eq.) of 3-azidopropionic acid succinimidyl ester (**3**) in 15 mL of absolute EtOH were added to the MSN suspension [46]. The mixture was stirred for 24 h at room temperature. Solid samples were collected by centrifugation at 13,000 rpm for 13 min, and washed and dispersed with EtOH six times. The solvent was evaporated under reduced pressure to give MSN–(N_3_).

### 2.7. Functionalization of (MSN–(N_3_)) with 4-(n-Butylamino)-N-(2-Propargyl)-1,8-Naphthalimide (MSN–(TNaph))

Twenty-six milligrams of MSN–(N_3_) were suspended in 20 mL of ACN. Then 7.7 mg of 4-(*n*-butylamino)-*N*-(2-propargyl)-1,8-naphthalimide (**4**) (0.025 mmol, 2 eq.) with 3 mg CuI (0.001 mmol) and three drops of DIPEA were added [47]. The mixture was stirred for 48 h at room temperature in the dark. Solid samples were collected by centrifugation at 13,000 rpm for 13 min and washed with diethyldithiocarbamate until the yellow color disappeared. Twelve washings were needed in order to remove the Cu–diethyldithiocarbamate complex (monitored by UV–vis at 730 nm) [35,37]. Afterward, MSNs were suspended in a methanolic solution of HCl 0.1 M/MeOH for 12 h. Finally, solid samples were collected by centrifugation and washed and dispersed in EtOH.

### 2.8. Synthesis of Bifunctionalized Amino-Isothiocyanate MSNs (MSN–(NH_2_)_i_(NCS)_o_)

Forty-four milligrams of MSN–NH_2_ (CTAB) were dissolved in 20 mL of dry toluene and stirred at 50 °C for 24 h. Then, 42 mg of thiocarbonyldi-2(1*H*)-pyridone (**1**) (0.180 mmol, 20 eq.) were added and the mixture was stirred at room temperature for 24 h. Solid samples were collected by centrifugation at 13,000 rpm for 13 min and then washed with absolute EtOH twice. Then, the tensioactive component was eliminated by adding to the dispersion 30 mL of a 0.1 M NH_4_NO_3_ methanolic solution for 24 h. Solid samples were collected by centrifugation and then washed and dispersed with absolute EtOH and water. The solvent was evaporated under reduced pressure and the resulting MSN–(NH_2_)_i_(NCS)_o_ were stored dry.

### 2.9. Ataluren Loading in MSN–(NH_2_)_i_(NCS)_o_)

One hundred milligrams of MSN–(NH_2_)_i_(NCS)_o_ were dissolved in 75 mL of absolute ethanol and stirred vigorously at room temperature for 1 h 30 min. Then, 67 mg of Ataluren (**5**) (0.236 mmol) were added and the mixture was stirred at room temperature for 24 h. Finally, solid samples were collected by centrifugation at 13,000 rpm for 13 min. Nanoparticles were washed with ethanol once.

### 2.10. Synthesis of Tert-Butyl(2-((2-Isothiocyanatoethyl)Disulfanyl)Ethyl)Carbamate (**6**) 

0.79 g (3.11 mmol) of *tert*-butyl(2-((2-aminoethyl)disulfanyl)ethyl)carbamate synthesized from cystamine dihydrochloride [48] and 0.68 g (2.95 mmol) of 1,1′-thiocarbonylbis(pyridine-2(1*H*)-one) (**1**) were dissolved in 50 mL of anhydrous DCM. The reaction was stirred at 25 °C for 24 h. Then, the organic phase was washed with H_2_O and NaHCO_3_ twice. Afterwards, the organic layer was dried over MgSO_4_ and then the solution was evaporated. Yield: 0.057 g (ƞ = 5.7%).

**^1^H NMR** (400 MHz, CDCl_3_) δ 4.85 (s, 1H), 3.84 (t, *J* = 6.6 Hz, 2H), 3.46 (q, *J* = 6.3 Hz, 2H), 2.94 (t, *J* = 6.6 Hz, 2H), 2.82 (t, *J* = 6.4 Hz, 2H), 1.45 (s, 9H). **^13^C NMR** (100 MHz, CDCl_3_) δ 155.7, 132.8, 79.7, 44.0, 39.3, 38.5, 37.9, 28.4. **IR (KBr) ν_max_**: 3354.18, 2976.65, 2929.07, 2187.71, 2113.20, 2082.62, 1699.95, 1509.44 cm^−1^. **Calculated organic elemental analysis (OEA)** C: 40.79%; H: 6.16%; N: 9.51%, O: 10.87%, S: 32.66%. **Experimental OEA** C: 40.71%; H: 6.37%; N: 9.31%; S: 32.15%.

### 2.11. Synthesis of Tert-Butyl (51-Thioxo-2,5,8,11,14,17,20,23,26,29,32,35,38,41,44,47-Hexadecaoxa-55,56,Dithia-50,52-Diazaoctapentacontan-58-yl)Carbamate (**7**)

1.73 mmol of 2,5,11,14,17,20,23,26,29,32,35,38,41,44,47-hexadecaoxanonatetraconta-49-amine synthesized from 2,5,11,14,17,20,23,26,29,32,35,38,41,44,47-hexadecaoxanonatetraconta-49-ol [49] and 562 mg (1.91 mmol) of **6** were dissolved in 50 mL of anhydrous DCM. The reaction was stirred at 20 °C for 24 h. The resulting product **7** was purified by silica gel column chromatography using a DCM:ACOEt mixture (50:50). The solvent was removed under vacuum to yield the pure product **7**. Yield: 0.236 g (ƞ = 27%).

**^1^H-NMR** (400 MHz, CDCl_3_) δ 6.94 (brs, 1H), 5.24 (brs, 1H), 3.90 (brs, 2H), 3.74–355 (overlapping multiplets, 72H), 3.45 (brs, 2H), 3.37 (s, 3H), 2.96 (brs, 2H), 2.78 (brs, 2H), 1.45 (s, 9H). **^13^C-NMR** (100 MHz, CDCl_3_) δ 183.2, 155.9, 79.5, 71.8, 70.5, 70.4, 70.2, 70.0, 58.9, 44.0, 39.4, 37.9, 37.8, 28.4. **IR (KBr) ν_max_**: 3517, 3335, 2871, 1709, 1542, 1454, 1107 cm^−1^
**MALDI-TOF** Matrix: DHB 10 mg/mL acetone, Ratio Sample–Matrix–NaTFA: 4:20:2, Method: Reflector positive ion mode (RP_master): 619.916, 692.848, 745.783, 736.931, 758.927, 878.043, 922.075, 965.111, 1009.188, 1053.275, 1053.275, 1097.309, 1141.357, 1185.463, 1229.516, 1273.535.

### 2.12. Synthesis of 1-(2-((2-Aminoethyl)Disulfanyl)Ethyl)-3-(2,5,8,11,14,1,20,23,26,29,32,35,38,41,44,47-Hexadecaoxanonatetracontan-49-yl)Thiourea (**8**)

Ten milliliters of anhydrous DCM were used to dissolve 0.26 mmol of **7**. The mixture was cooled at 0 °C and stirred for 30 min. Then, 1 mL (13 mmol) of trifluoroacetic acid (TFA) was added dropwise. Afterward, the mixture was stirred for 1.5 h at 0 °C and the solvent was removed under vacuum. Compound **8** was used without further purification.

### 2.13. Functionalization of MSN–(NH_2_)_i_(NCS)_o_ with Aminotetraethylene Glycol Monomethyl Ether and Loading of Ataluren (MSN–(NH_2_)_i_(PEG)_o_ (Ataluren))

Twenty-two milligrams of MSN–(NH_2_)_i_(NCS)_o_ were suspended in 30 mL of absolute EtOH and stirred vigorously for 1 h. Then, 16 mg of Ataluren (**5**) (0.056 mmol) were added to the solution. The mixture was stirred at room temperature for 24 h. Afterward, 30 mg (0.144 mmol, 181 eq.) of aminotetraethylene glycol monomethyl ether (PEG, **9**) was added, and the mixture was stirred for another 24 h at room temperature [50]. Solid samples were collected by centrifugation at 13,000 rpm for 13 min. Nanoparticles were washed with absolute ethanol twice. The washings were collected and the concentration of Ataluren was determined by UV–vis spectroscopy at 244 nm to calculate the loading of the drug by difference.

### 2.14. Functionalization of MSN–(NH_2_)_i_(NCS)_o_ with 1-(2-((2-Aminoethyl)Disulfanyl)Ethyl)-3-(2,5,8,11,14,1,20,23,26,29,32,35,38,41,44,47-Hexadecaoxanonatetracontan-49-yl)Thiourea and Loading of Ataluren (MSN–(NH_2_)_i_(SS-PEG)_o_ (Ataluren))

Thirteen milligrams of MSN–(NH_2_)_i_(NCS)_o_ were dissolved in 30 mL of absolute EtOH and stirred vigorously for 1 h. Then, 11 mg of Ataluren (**5**) (0.0397 mmol) were added to the solution. The mixture was stirred at 20 °C temperature for 24 h. Then, 216 mg (539.78 mmol) of **8** were added to the silica nanoparticles. Finally, the mixture was stirred for another 24 h. Solid samples were collected by centrifugation at 13,000 rpm for 13 min. Nanoparticles were washed with absolute EtOH twice. The washings were collected and the concentration of Ataluren was determined by UV–vis spectroscopy at 244 nm to calculate the loading of the drug by difference.

### 2.15. Functionalization of MSN–(NH_2_)_i_(NCS)_o_ with (2-Aminoethyl)Trimethylammonium Chloride Hydrochloride and Loading of Ataluren (MSN–(C^+^)_o_ (Ataluren))

Sixty milligrams of MSN–(NH_2_)_i_(NCS)_o_ were suspended in 30 mL of basic water (K_2_CO_3_ 0.2 M, pH = 8) and stirred vigorously for 1 h. Then, 50 mg of Ataluren (**5**) (0.176 mmol) were added to the solution. The mixture was stirred at room temperature for 24 h. Then, 7.6 mg of (2-aminoethyl)trimethylammonium chloride hydrochloride (**10**) (0.0436 mmol, 54.6 eq.) were added, and the mixture was stirred at room temperature for 24 h. Solid samples were collected by centrifugation at 13,000 rpm for 13 min. Nanoparticles were washed with absolute EtOH twice. The washings were collected and the concentration of Ataluren was determined by UV–vis spectroscopy at 244 nm to calculate the loading of the drug by difference.

### 2.16. Release Experiments

Ten milligrams of MSNs loaded with Ataluren were placed inside an Eppendorf. Then, 1.5 mL of citric acid 0.1M/sodium phosphate (Na_2_HPO_4_·12H_2_O) 0.2 M buffer (pH = 7.4) was poured into the latter. The Eppendorf was sonicated until a clear dispersion was obtained. After that, the solution was allowed to stand in an incubator for a specified time. The resulting precipitate was isolated by centrifugation at 13,000 rpm for 13 min. In order to calculate the Ataluren concentration through UV–vis spectroscopy at 244 nm, 1.3 mL of the supernatant was extracted and stored. The Eppendorf was refilled again with 1.3 mL of the previous buffer. All these operations were performed five more times (24 h of release). Ataluren release has been calculated using Equation (S1) (see Appendix A).

### 2.17. Characterization

A high-resolution transmission electron microscope (JEOL JEM 2011, Tokyo, Japan) was used to take micrographs of the MSNs particles. Samples were ultrasonically dispersed in absolute EtOH at a concentration of 0.1 mg·mL^−1^ and deposited on an amorphous, porous carbon grid. The particle size distribution and Z-potential were measured using a Malvern Zetasizer Nano Series ZEN 3600 laser particle size analyzer (Malvern, UK). Samples were prepared at a concentration of 0.1 mg·mL^−1^ in absolute ethanol for size and 0.05 mg·mL^−1^ in deionized H_2_O (pH = 5.5) for the Z-potential measurements. Small-angle powder X-ray diffraction (SAXRD) patterns were collected on a Philips X’Pert diffractometer (Malvern, UK) equipped with Cu-Kα radiation (wavelength 1.5406 Å) in the 2θ range between 0.6° and 6.5° with a step size of 0.02° and counting time of 5 s per step. Nitrogen physisorption analysis was conducted on a micromeritics Gemini V surface area and pore size analyzer (Norcross, GA, USA). Samples were treated at 0.05 mBar at −0.759 °C approximatively for 16 h prior to conducting adsorption experiments. The surface area was calculated by the Brunauer–Emmett–Teller (BET) method and the pore size distribution was calculated using the Barrett–Joyner–Halenda (BJH) method. Fourier transform infrared (FTIR) spectra of MSN particles were obtained via a Thermo Scientific, Waltham, MA, USA, Nicolet iS10 FTIR spectrometer with Smart iTr. A EuroVector Instruments, Euro EA elemental analyzer was used for the determination of elemental microanalyses. UV–vis spectra were recorded in a Thermo Scientific 300 UV–vis spectrophotometer. Fluorescence excitation spectra were recorded in a Hitachi F2500, Tokyo, Japan, fluorescence spectrophotometer.

## 3. Results and Discussion

### 3.1. Synthesis and Characterization of Isothiocyanate MSNs (MSN–(NCS))

From a practical point of view, the introduction of the isothiocyanate moiety should take place in one step by direct conversion of the amino groups into the isothiocyanate [51]. A search in the literature reveals that the typical procedure for the preparation of isothiocyanates from amines consists of a two-step methodology [52,53]. An alternative and straightforward protocol is the use of the “thiocarbonyl transfer reagents” [54] such as 1,1′-thiocarbonyldi-2(1*H*)-pyridone (**1**). This commercially available reagent reacts smoothly with primary amines under neutral conditions to give high yields of the corresponding isothiocyanate in a single step. The only by-product formed is the water-soluble 2-pyridone [55].

As proof of concept of this chemistry, aminated MSNs (50 and 100 nm), prepared following the methodology described by Lo et al. [44], were reacted with 12 eq. of 1,1′-thiocarbonyldi-2(1*H*)-pyridone (**1**) in toluene for 24 h (Figure 1). The resulting MSNs show good chemical stability and can be stored at room temperature indefinitely (checked by reaction of MSN–(NCS) with amine **2**).

The nanoparticles were characterized by dynamic light scattering (DLS), powder the small-angle X-ray diffraction (SAXRD), TEM, and BET analysis (see Appendix A). As expected, no significant size, shape, and morphology differences were obtained for MSN–(NCS) in comparison with aminated-MSNs. MSN–(NCS) are regular, homogeneous and round shaped. As shown in Figure 2, spherical 100 nm nanoparticles with typical mesoporous morphology were obtained. The particle size distribution was measured by dynamic light scattering (DLS) as shown in Appendix A, with the average particle size of 142 (pdI = 0.07) and 190 nm (pdI = 0.19) for MSN–(NH_2_) and MSN–(NCS), respectively. Z-potential was also measured. The values recorded were −12 and −11 mV for 100 nm MSN–(NH_2_) and MSN–(NCS), respectively (As for the 50 nm nanoparticles, see Appendix A).

Powder SAXRD patterns of MSN–(NCS) was also carried out. SAXRD pattern presents highly ordered structures with d_100_ at 2.3 and lighter faceted hexagon-shape at 4.1 and 4.2 which indicated two-dimensional (2D) long-range ordering structure (Figure 3).

The nitrogen adsorption–desorption technique was used to analyze the surface area, pore volume, and pore size distribution of the MSN–(NH_2_) and MSN–(NCS) particles. BET isotherm curves of MSN samples (100 nm nanoparticles), as displayed in Figure 4, showed type IV isotherms, which indicate clear H1 hysteresis loop characteristic of mesoporous materials. BET surface areas are over 1120 m^2^·g^−1^ for MSN–(NH_2_) and 850 m^2^·g^−1^ for MSN–(NCS). Additionally, the pore volume for MSN–(NCS) was 0.53 cm^3^·g^−1^. As a reference, the value recorded for MSN–(NH_2_) was 0.72 cm^3^·g^−1^. These decreases in both the surface area and the pore volume are consistent with the functionalization of the pores of the nanoparticle. The MSN–(NCS) present a very narrow pore size distribution centered at 2.2 nm. These data suggest that the mild conditions used for the functionalization do not erode the structural features of the MSNs (As for the 50 nm nanoparticles, see Appendix A).

Furthermore, the successful functionalization of the MSNs is supported by the presence of two characteristic absorption bands around 2100 cm^−1^ in the FTIR spectrum ascribed to the isothiocyanate group (Figure 5).

### 3.2. Assessment of the Functionalization of MSN–(NCS)

In order to test the functionalization of these new isothiocyanate MSNs, a fluorescent naphthalimide bearing an aliphatic primary amine was chosen. Then, 4-(*n*-butylamino)-*N*-(2-aminoethyl)-1,8-naphthalimide (**2**) was synthetized from 1,8-bromonaphthalimide following a literature procedure [45]. Thus, the isothiocyanate-functionalized MSNs were exposed to the primary amine for 48 h using dichloromethane as a solvent to form MSN–(UNaph) (Appendix A). The reacting mixture was washed with the same solvent and centrifuged to render a yellow solid. The disappearance of the two bands around 2100 cm^−1^ (isothiocyanate group) indicates the completion of the reaction and the formation of the corresponding thiourea (Appendix A).

To assess the performance of this new functionalization methodology, the CuAAC coupling was used as a standard [35,36]. Azido-(MSNs) were synthetized by functionalizing 3-azidopropionic acid succinimidyl ester (**3**) to the initial aminated MSNs (Appendix A). Furthermore, 3-azidopropionic acid succinimidyl ester (**3**) was synthetized from 3-bromopropionic acid following the literature [46]. Therefore, the corresponding azido-functionalized MSNs were coupled with a homologous napththalimide attached to a terminal alkyne to form MSN–(TNaph). Moreover, 4-(*n*-butylamino)-*N*-(2-propargyl)-1,8-naphthalimide (**4**) was synthetized from 1,8-bromonaphthalimide following the literature [47]. The disappearance of the azido band around 2100 cm^−1^ indicated the completion of the reaction (Appendix A).

Both MSN–(NCS) and MSN–(N_3_) nanoparticles were successfully functionalized with naphthalimide moieties presenting yellow coloration and a maximum band in the absorption spectrum of approximatively 450 nm (Appendix A). The functionalization capacity of both azido- and isothiocyanate MSNs (MSN–(NCS)) and (MSN–(N_3_)) was determined by organic elemental analysis (OEA) (Table 1). To our delight, the loading turns out to be comparable in both cases (4.6% when the CuAAC protocol was used, whereas the value was 4.8% for the new methodology).

As anticipated, these results confirm the suitability of the isothiocyanate group to participate in the functionalization of MSNs. The main difference in the protocols is the isolation of the nanoparticles. While the thiourea-containing MSNs are easily isolated, the removal of copper species from the CuAAC reaction requires tedious and extensive washings with solutions of chelating agents such as Na_2_EDTA [37] or *N*,*N*-diethyldithiocarbamate [35].

### 3.3. Synthesis and Characterization of Bifunctionalized Amino-Isothiocianate MSNs (MSN–(NH_2_)_i_(NCS)_o_)

Having validated the performance of the new isothiocyanate mono-functionalized MSNs, we were prompted to prepare regioselectively bifunctionalized amino-isothiocyanate MSNs adapting the previous protocol. Briefly, aminated MSNs (50 and 100 nm) [44] containing the surfactant (CTAB) inside the pore were reacted with 12 eq. of 1,1′-thiocarbonyldi-2(1*H*)-pyridone (**1**) in dry toluene for 24 h (Figure 6). This solvent was carefully chosen in order to prevent the elimination of the surfactant. Then, the resulting solid was washed once with absolute ethanol and dry toluene.

At this stage, the FTIR spectrum of the MSNs confirms the successful introduction of the isothiocyanate moiety (absorption bands around 2100 cm^−1^) and the presence of the surfactant blocking the pore, which can be inferred from a typical absorption at 2990 cm^−1^ (C–H stretch) (Figure 5).

Additional proof of the occlusion of the pores is given by the BHJ porous size distribution (Figure 7a and Appendix A and Table 2 and Appendix A). Hence, the functionalization of the MSNs was carried out regioselectively in the outer surface thanks to the smooth conversion of amine groups into isothiocyanate by reagent (**1**), which does not disturb the surfactant.

Finally, the surfactant was removed by treatment of the nanoparticles in a mixture NH_4_NO_3_/methanol (treatment with HCl/methanol gives similar results). The complete removal of the template can be confirmed by comparing the change in the specific surface (Figure 7b).

The nanoparticles were characterized by the standard techniques. Again, as expected, no significant size, shape, and morphology differences were obtained for MSN–(NH_2_)_i_(NCS)_o_ in comparison with aminated MSNs. MSN–(NH_2_)_i_(NCS)_o_ are regular, homogeneous, and round shaped. As shown in Figure 8, spherical 100 nm nanoparticles with typical mesoporous morphology were obtained.

The particle size distribution was measured by dynamic light scattering (DLS) as shown in Appendix A (100 nm), with the average particle size of 142 (pdI = 0.07) and 173 nm (pdI = 0.04) for MSN–(NH_2_) and MSN–(NH_2_)_i_(NCS)_o_, respectively. Z-potential was also measured with −12 and −13 mV, respectively. Powder XRD patterns of MSN–(NCS) was also carried out. The small-angle *X*-ray diffraction (SAXRD) pattern presents highly ordered structures with d100 at 2.3 and lighter faceted hexagon-shape at 4.1 and 4.2 which indicated two-dimensional (2D) long-range ordering structure (Figure 3).

The N_2_ adsorption/desorption measurements for MSN–(NH_2_)_i_(NCS)_o_ showed type IV isotherms, which display clear H1 hysteresis loop characteristic of mesoporous materials (Figure 7a). For 100 nm nanoparticles, BET surface areas were over 1120 m^2^·g^−1^ for MSN–(NH_2_), whereas those for MSN– (NH_2_)_i_(NCS)_o_ were 1000 m^2^·g^−1^ Additionally, the pore volume for MSN–(NH_2_)_i_(NCS)_o_ was 0.63 cm^3^·g^−1^. As a reference, the value recorded for MSN–(NH_2_) was 0.72 cm^3^·g^−1^. The MSNs present a very narrow pore size distribution centered at 2.2 nm (Table 2) (As for the 50 nm nanoparticles, see Appendix A). Again, these nanoparticles are characterized by an excellent chemical stability and can be kept for about three months at room temperature (23 °C).

### 3.4. Application of MSN–(NH_2_)_i_(NCS)_o_ for the Preparation of a Nanocarrier for Ataluren Release

The chemical stability of the MSN–(NH_2_)_i_(NCS)_o_ along with the clean reactivity and easy purification of the particles endows these systems with ideal properties to be used in the design of drug carriers as ready-to-use building blocks. The isothiocyanate groups located in the surface of the MSNs are amenable to react with a large variety of primary amines, ranging from simple alkyl amines, short PEGs to polymers. Once attached to the MSNs, their presence can modulate the release profile of the payload [13,14,15,16]. The chemical nature of these chains is a key factor to optimize not only the drug release but also the characteristics of the corona of the nanoparticles which define their biodistribution and clearance from the body [56]. Hence, a short and simple protocol to install such chemical valves from a large variety of commercially available amines can be of broad applicability.

As an application of this new methodology, a very simple nanocarrier for the delivery of Ataluren (Translarna™) [57] was designed. To this end, bifunctionalized amino-isothiocyanate nanoparticles (50 nm) were prepared (Figure 6). The presence of the amino groups in the inner surface of the porous is a key feature of the design, since it has been shown that amino groups enhance the loading and facilitates the release of carboxylic acid containing drugs [13,14,15,16].

Briefly, bifunctionalized MSN–(NH_2_)_i_(NCS)_o_ nanoparticles were loaded with the drug by exposing them to a solution of Ataluren in absolute ethanol or water as solvents. Then, the nanoparticles were reacted with a primary amine to seal the pores. Finally, the mixture was centrifuged and washed twice with ethanol. It is noteworthy that the only product present in the washing was the slight excess of the primary amine. Three amines were chosen: 1-(2-((2-aminoethyl)disulfanyl)ethyl)-3-(2,5,8,11,14,1,20,23,26,29,32,35,38,41,44,47-hexadecaoxanonatetracontan-49-yl)thiourea (**8**), 3,6,9,12-Tetraoxatridecan-1-amine (**9**) and (2-aminoethyl)trimethylammonium chloride (**10**) (Figure 9) [55].

The blocking effect of the amino-PEG (**9**) was clearly demonstrated by the release profile of Ataluren (Figure 10). Whereas for non-functionalized MSNs, Ataluren release was 52.7%, for polyethylene glycol (**9**) MSNs (MSN–(NH_2_)_i_(PEG)_o_ (Ataluren)), the release was just 7.9%, which clearly demonstrates the blocking effect of the PEG chain. A better obstruction of the pores can be achieved if the MSN is functionalized with a longer polyethylene glycol chain (*n* = 15) (**8**) (MSN–(NH_2_)_i_(SS-PEG)_o_ (Ataluren)) where only 4.1% of Ataluren was released. In the case of glycol (**8**), the PEG chain contains a labile disulfide bond, which under reductive conditions can be cleaved releasing the cargo, giving 22% of Ataluren (in a period of 6 h).

However, since a slow and progressive release of the drug is recommended for its therapeutic use [58,59], a shorter amine was chosen. Thus, the PEG amine was substituted by (2-aminoethyl)trimethylammonium chloride (**10**). The aim of using a quaternized amine was to slow down the release of Ataluren by means of ionic interactions between the drug and the quaternary amine moiety. In this case, a softer and more controlled release of Ataluren was obtained (Figure 10). While for non-functionalized MSNs complete Ataluren release was obtained at 2 h, for MSN–(NH_2_)_i_(C^+^)_o_(Ataluren) the release profile was smoother providing 25.5% of Ataluren in a period of 6 h.

## 4. Conclusions

In conclusion, a straightforward protocol to prepare isothiocyanate-functionalized MSNs has been described. The synthetic methodology is general and can be applied, in principle, to all types of aminated MSNs. The resulting MSNs are chemically stable and can be stored indefinitely. As anticipated, the particles easily react with primary amines and they are compatible with aqueous media. The efficiency of the functionalization is comparable to that of CuAAC. However, in stark contrast with the CuACC protocols, the isolation of the derivatized MSNs is simple and there is no need to remove any by-product or toxic catalysts. Furthermore, this methodology was applied to the regioselective synthesis of amino-isothiocyanate-functionalized MSNs. These nanoparticles have been used for the design of a nanocontainer able to release the drug Ataluren. The release profile of the drug can be fine-tuned with the careful choice of the capping amine. The application of such nanoparticles in the development of “smart” nanocarriers is currently ongoing in our laboratories.

## Figures and Tables

**Figure 1 nanomaterials-09-01219-f001:**
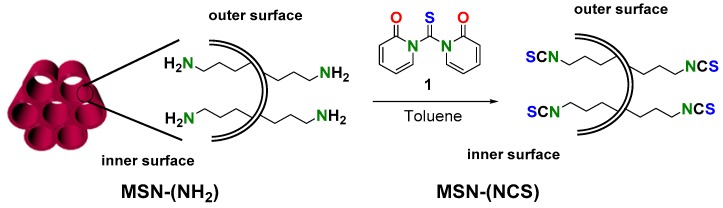
Preparation of mesoporous silica nanoparticle (MSN)–(NCS).

**Figure 2 nanomaterials-09-01219-f002:**
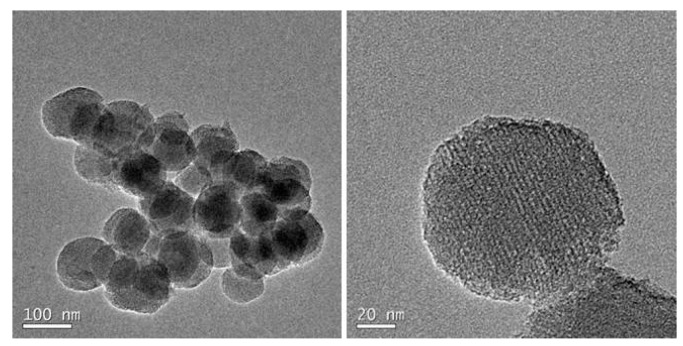
Representative transmission electron microscope (TEM) images of 100 nm MSN–(NCS).

**Figure 3 nanomaterials-09-01219-f003:**
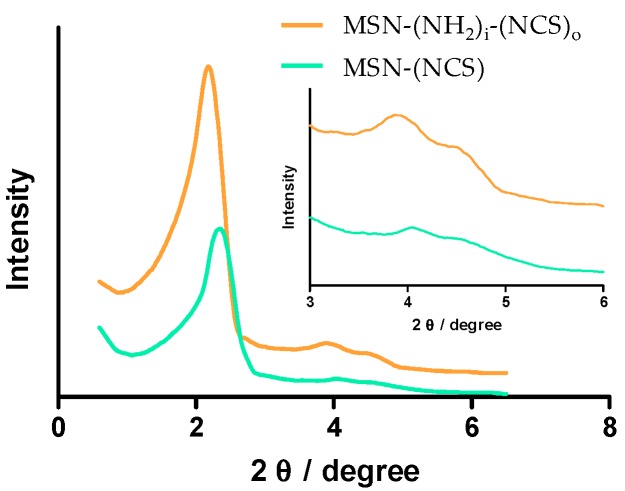
Small-angle X-ray diffraction (SAXRD) of MSN–(NCS) and MSN–(NH_2_)_i_(NCS)_o_ (100 nm).

**Figure 4 nanomaterials-09-01219-f004:**
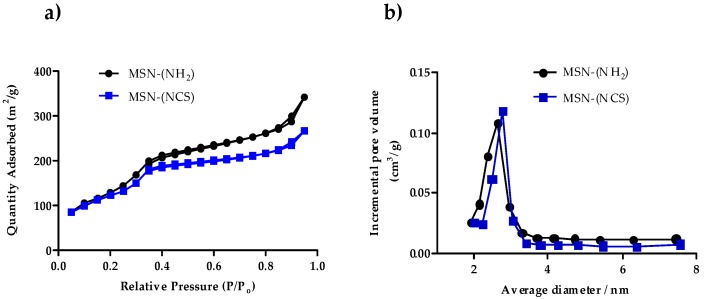
(**a**) Nitrogen adsorption–desorption isotherms of MSN–(NH_2_) and MSN–(NCS), (**b**) pore size distribution of MSN–(NH_2_) and MSN–(NCS) (100 nm).

**Figure 5 nanomaterials-09-01219-f005:**
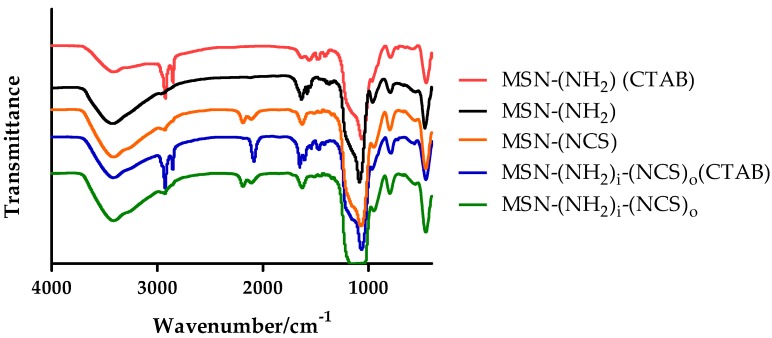
FTIR spectra of MSN–(NH_2_) (CTAB), MSN–(NH_2_), MSN-(NCS), MSN–(NH_2_)_i_(NCS)_o_ (CTAB), and MSN–(NH_2_)_i_(NCS)_o_.

**Figure 6 nanomaterials-09-01219-f006:**
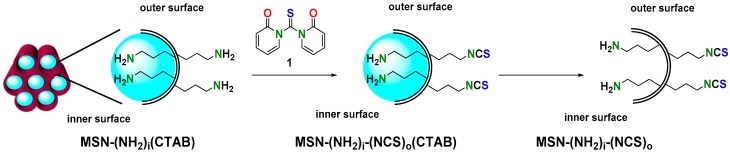
Regioselective synthesis of MSN–(NH_2_)_i_(NCS)_o_.

**Figure 7 nanomaterials-09-01219-f007:**
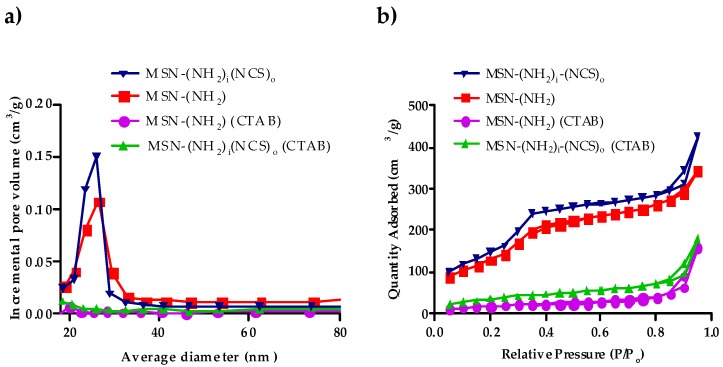
Characterization of regioselectively functionalized MSN–(NH_2_)_i_(NCS)_o_. (**a**) Pore size distribution, (**b**) nitrogen adsorption–desorption isotherms (100 nm).

**Figure 8 nanomaterials-09-01219-f008:**
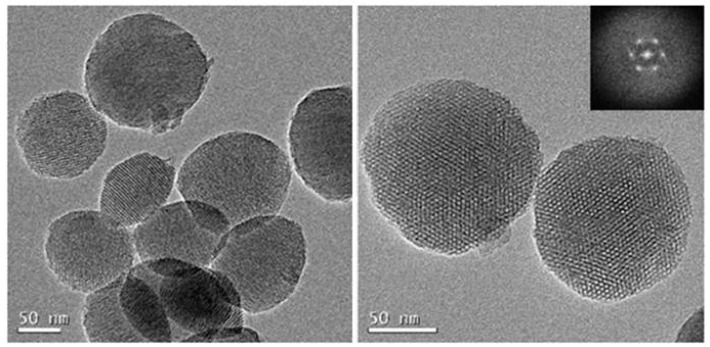
TEM images of MSN–(NH_2_)_i_(NCS)_o_ (100 nm).

**Figure 9 nanomaterials-09-01219-f009:**
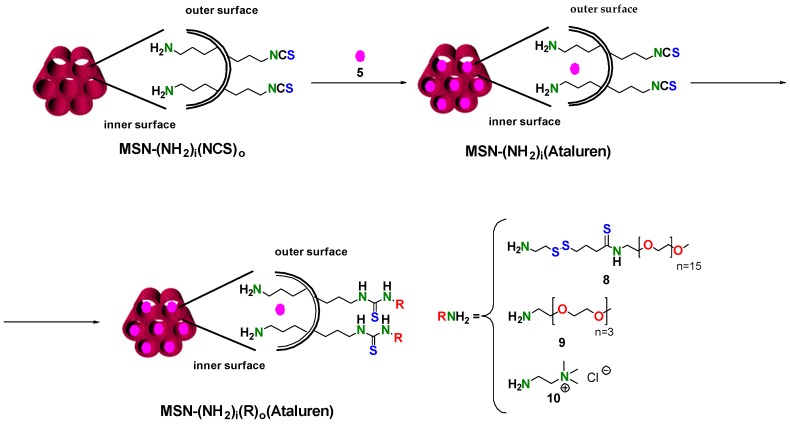
Loading of MSN–(NH_2_)_i_(NCS)_o_ with Ataluren and functionalization with 1-(2-((2-aminoethyl)disulfanyl)ethyl)-3-(2,5,8,11,14,1,20,23,26,29,32,35,38,41,44,47-hexadecaoxanonatetracontan-49-yl)thiourea (**8**), 3,6,9,12-tetraoxatridecan-1-amine (**9**) and (2-aminoethyl)trimethylammonium chloride (**10**).

**Figure 10 nanomaterials-09-01219-f010:**
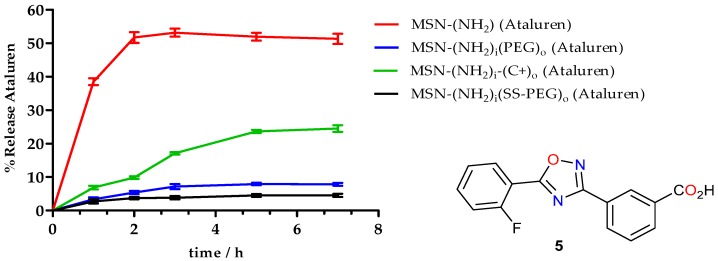
Release profile of Ataluren: Aminated nanoparticles (MSN–(NH_2_) (Ataluren)), aminated nanoparticles functionalized with (2-aminoethyl)trimethylammonium (**10**) (MSN–(NH_2_)_i_(C^+^)_o_ (Ataluren)), aminated nanoparticles functionalized with polyethylene glycol (**9**) (MSN–(NH_2_)_i_(PEG)_o_ (Ataluren)), and aminated nanoparticles functionalized with polyethylene glycol disulfide bond (**8**) (MSN–(NH_2_)_i_(SS-PEG)_o_ (Ataluren)).

**Table 1 nanomaterials-09-01219-t001:** Organic elemental analysis (OEA) of MSN–(NCS), MSN–(UNaph), MSN–(N_3_), and MSN–(TNaph).

MSNs	C (%)	H (%)	N (%)
MSN–(NCS)	9.78	1.87	1.92
MSN–(UNaph)	14.59	2.59	2.32
MSN–(N_3_)	7.22	2.56	2.64
MSN–(TNaph)	11.79	2.56	3.08

**Table 2 nanomaterials-09-01219-t002:** N_2_ adsorption–desorption (BET) and Barrett–Joyner–Halenda (BJH) pore size distribution values of MSN–(NH_2_) (CTAB), MSN–(NH_2_), and MSN–(NH_2_)_i_(NCS)_o_ of 100 nm.

Property	MSN–(NH_2_) (CTAB)	MSN–(NH_2_)	MSN–(NH_2_)_i_(NCS)_o_
BET surface area (m^2^/g)	17.3	1120.90	1000.70
BJH pore volume (cm^3^/g)	0.03	0.72	0.63
Pore size (nm)	–	2.20	2.20

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
