# Peer review of "Isothiocyanate-Functionalized Mesoporous Silica Nanoparticles as Building Blocks for the Design of Nanovehicles with Optimized Drug Release Profile"

_nanomaterials, 2019, doi:10.3390/nano9091219_

Round 1
Reviewer 1 Report
Line 34 : replace “it is necessary the development” by it is necessary to develop
Line 72 : replace triethoxisilane by triethoxysilane
2.1 Materials : several chemical products are missing in the list : HCl, Toluene, Ethanol and so on… Please update the list of chemicals.
Line 87 : please, specify the concentration of HCl solution which is mixed with Ethanol . Indicate what kind of Ethanol was used (absolute, 95%v/v).
Line 89 : Assign properly the specific band of CTAB.
Line 90 : the method to quantify the number of NH2/mg of MSN should be described in detail.
Line 93 : Specify what kind of toluene is used for the synthesis of MSN-(NCS). Absolute ?
Line 97 : the volumes of ethanol and toluene are not specified.
Line 104 : the author should explain how they controlled the disappearance of the yellow color.
Centrifugation conditions are not always given. Please specify each time your conditions.
Paragraph 2.4 and 2.8 : the synthesis are carried out in toluene at two different temperatures : 50°C and 60°C. Is there any reason for that ?
Line 101 : Please specify the degree of purity of ethanol
Line 133 : the conditions of centrifugation are uncomplete.
Line195 : The pH buffer should be properly described. What is the expected pH ? 0.2M phosphate does not exist. Please specify the kind of phosphate you used.
Line 204 : why changing the solvent to do size and zeta potential measurements ?
Figure 3 : the quality of the figure is too low. The legend of figure 3 does not match with the name of the samples in the text.
Figure 3 : A zoom of the second and third peaks between 4 and 5° should be added in the figure.
Figure 5 : the quality of the figure is too low.
Line 274 : the authors do not explain why the disappearance the band at 2100 cm-1 support the completion of the reaction.
Line 301 and 308: change porous by pore
Figure 7 : the IR spectra of MSN_NH2 should be added in Figure 7
Paragraph 3.3 : The authors should explain why the figure 8 clearly demonstrates that CTAB remained in the pores. Same comment concerning Figure 8a. Besides, we usually cite Figure 8a before Figure 8b and not the opposite.
Line 321 : the surfactant was removed by refluxing in HCl/EtOH or NH4NO3/methanol. This step should be described properly in the experimental part of the paper. Furthermore, the authors did not explain why they tried two strategies to remove the surfactant and whether one strategy is better than the other.
Suggestion to the authors : a table with the ssa and pore volume would be more helpful for the clarity of the manuscript than the curves in Figure 8.
Line 369 : three and not there
Questions to the authors :
Paragraph 2.16 : the release experiments should be described in detail. For example, the author should explain how they have withdrawn Ataluren without withdrawing MSN loaded with Ataluren? Line 229 : what « good chemical stability » means ? What kind of test did you use to check the stability ? Line 238 : how can you explain such a size difference between MSN-NH2 and MSN-NCS. The PDIs are also strongly different suggesting agglomeration due to the surface modification. Thank you for your comments. Line 239 : Could you comment on the negatively charged MSN-NH2 ? We could expect either neutral charged or positively charged nanoparticles due to the presence of amine groups on the surface of MSN. Figure 3 : it seems that the main peaks is slightly shifted to larger values of 2 theta after the grafting. Can you comment on that ? Besides in the table S1, we can see that the DLS size is much higher that the TEM size especially when the particles are 50 nm in diameter. Can you explain such a difference ? Line 253 : The authors should explain the decrease of the sorption propoerties of MSNs after grafting the isothiocyanate moiety : decrease of ssa and pore volume, while the pore size remains constant. Line 263 : The authors should better describe the IR spectra and especially the double band 2100 cm-1. Besides, the different IR curves do not correspond to MSN-NH2 or MSN-NCS. Why ? Paragraph 3.3 : The authors should explain why the figure 8 clearly demonstrates that CTAB remained in the pores. Same comment concerning Figure 8a. Besides, we usually cite Figure 8a before Figure 8b and not the opposite. Line 346 : What the author mean by “indefinitely”. How long time did the author let the MSNs at room temperature before checking that the porous properties of the materials did not change? Besides, what do you mean by room temperature? In ambient air? Figure 11 : the authors have to explain the term %release Ataluren. How did they measure the total amount of drug in the particles ? The author claim that the drug is released faster in the presence of a shorter PEG chain. Did the author try to determine the amount of PEG grafted with the different precursors? The information would be of great importance to better conclude on the influence of the chain length on the drug release even if we can easily expect that short chains will close the pores less easily that large chains. Can the author comment on the slower drug release in the presence of molecule 10 than with molecule 9 whereas molecule 09 is bigger?Author Response
Referee comments nb 1:
Comments and Suggestions for Authors
Line 34 : replace “it is necessary the development” by it is necessary to develop
Changed in the text as requested.
Line 72 : replace triethoxisilane by triethoxysilane
Changed in the text as requested.
2.1 Materials : several chemical products are missing in the list : HCl, Toluene, Ethanol and so on… Please update the list of chemicals.
We thank the reviewer for such comment. All the chemicals have been added in the text as requested:
“Cetyltrimethylammonium bromide (CTAB), tetraethylorthosilicate (TEOS), 3-aminopropyl triethoxysilane (APTES), dry toluene, absolute ethanol, methanol, acetonitrile (ACN), anhydrous dichloromethane (DCM), ethyl acetate (EtOAc) and tetraethylene glycol monomethyl ether were purchased from ACROS; Ammonium hydroxide (NH4OH) from Fluka; 1,1′-Thiocarbonyldi-2(1H)-pyridone, (2-aminoethyl)trimethylammonium chloride hydrochloride, hydrochloric acid (HCl), trifluoroacetic acid (TFA), sodium bicarbonate (NaHCO3), magnesium sulfate (MgSO4), copper iodide (CuI), N,N-Diisopropylethylamine (DIPEA), sodium diethyldithiocarbamate, ammonium nitrate (NH4NO3), potassium carbonate (K2CO3), cystamine dihydrochloride, Ataluren, citric acid, sodium phosphate (Na2HPO4·12H2O), 3-bromopropionic acid, 2,5,8,11,14,17,20,23,26,29,32,35,38,41,44,47-hexadecaoxanonatetracontan-49-ol Fluorescein-5-isothiocyanate (FITC) and 1,8-bromonaphthalimide from Sigma Aldrich. All the chemicals were used as received without further purification”
Line 87 : please, specify the concentration of HCl solution which is mixed with Ethanol . Indicate what kind of Ethanol was used (absolute, 95%v/v).
Both specifications have been introduced in the text as requested.
Line 89 : Assign properly the specific band of CTAB.
The second band of CTAB has been added as requested.
Sentence added as requested: “bands disappearance at 2900 cm-1 and 2800 cm -1”
Line 90 : the method to quantify the number of NH2/mg of MSN should be described in detail.
The following methodology has been added to the text as requested:
“To a solution of 10 mg of MSN-(NH2)CTAB in 10 mL of dry toluene were added 10 mg of FTIC. This solution was stirred 24 h at 23 °C. Solid samples were collected by centrifugation at 13000 rpm for 13 min and the supernatant was measured at 495 nm to determine the quantity of unreacted FTIC. The difference with the total amount of FITC added gave the amino groups present onto the external surface of MSN-(NH2)CTAB. The same procedure was repeated with MSN-NH2, without the surfactant, to determine the total amount of amino groups present onto the inner and outer surface. Therefore, in total, there were approximately 8·10-7 NH2 mol/mg MSN, from which 6·10-7 NH2 mol/mg MSN onto the inner surface of the MSN.”
Line 93 : Specify what kind of toluene is used for the synthesis of MSN-(NCS). Absolute ?
Specified in the text as requested.
Line 97 : the volumes of ethanol and toluene are not specified.
Specified in the text as requested.
Line 104 : the author should explain how they controlled the disappearance of the yellow color.
Explained in the text as requested.
L 118: “until disappearance of yellow color (5 times, confirmed by tlc of the supernatant)”.
A tlc was run with the supernatants to confirm that 5 washings were enough.
Centrifugation conditions are not always given. Please specify each time your conditions.
All centrifugations conditions have been standardized and specified as requested.
Conditions added as requested: “centrifugation at 1300 rpm for 13 min”
Paragraph 2.4 and 2.8 : the synthesis are carried out in toluene at two different temperatures : 50°C and 60°C. Is there any reason for that ?
We thank the referee for this appreciation. Both temperatures must be the same 50 °C, hence it has been changed in the text.
Line 101 : Please specify the degree of purity of ethanol
Specified as requested
“Absolute ethanol”
Line 133 : the conditions of centrifugation are uncomplete.
All the centrifugation conditions have been updated and complete as requested (“13000 rpm for 13 minutes”).
Line195 : The pH buffer should be properly described. What is the expected pH ? 0.2M phosphate does not exist. Please specify the kind of phosphate you used.
The referee is correct. The pH of citric acid 0.1M / sodium phosphate (Na2HPO4·12H2O) 0.2 M buffer is 7.4.
The following sentence has been added:
L 212: “citric acid 0.1M / sodium phosphate (Na2HPO4·12H2O) 0.2 M buffer (pH=7.4)”
Line 204 : why changing the solvent to do size and zeta potential measurements ?
In order to obtain robust results, it is essential to determine a suitable concentration of nanoparticles and a proper solvent for each type of measurement. Thus, after several assays, it was found that, the best measurements were obtained for concentrations of 0.1 mg MSN/mL (EtOH) and 0.05 mg/mL (deionized H2O) to determine size and zeta potential, respectively.
Figure 3 : the quality of the figure is too low. The legend of figure 3 does not match with the name of the samples in the text.
A better-quality image has been added in the text and the figure name has been corrected.
Figure 3 : A zoom of the second and third peaks between 4 and 5° should be added in the figure.
A zoom for the requested section has been added in figure 3.
Figure 5 : the quality of the figure is too low.
A better-quality figure has been added in the text.
Line 274 : the authors do not explain why the disappearance the band at 2100 cm-1 support the completion of the reaction.
The band around 2100 cm-1 is ascribed to the isothiocyanate group. When MSN-NCS react with the amino group of of 4-(n-butylamino)-N-(2-aminoehtyl)-1,8-naphthalimide a thiourea is obtained. Hence, in the absence of any other functional groups, it is understood that the reaction has taken place. Further support to the successful functionalization is given by UV-vis spectroscopy. A characteristic band of the chromophore emerges around 450 nm (Fig. S7. Absorption spectra of MSN-(UNaph))
The following sentence has been added:
L299: “The disappearance of the two bands around 2100 cm-1 (isothiocyanate group) indicates the completion of the reaction and the formation of the corresponding thiourea”
Line 301 and 308: change porous by pore
Changed as requested.
Figure 7 : the IR spectra of MSN_NH2 should be added in Figure 7
The spectrum is added as requested (figure 5)
Paragraph 3.3 : The authors should explain why the figure 8 clearly demonstrates that CTAB remained in the pores. Same comment concerning Figure 8a. Besides, we usually cite Figure 8a before Figure 8b and not the opposite.
As CTAB remains in the pores, N2 is not able to penetrate through. In consequence, the surface area and the pore volume are extremely small. In contrast, once the CTAB is removed a typical IV isotherm type is obtained and a porous size distribution of 2.2 nm is found. In addition, FTIR spectrum is consistent with the presence of the surfactant (Figure 7).
Probably, it is clearer to mention the value of the pore size given in Table 2:
L 339: “Additional proof of the occlusion of the porous is given by the BHJ porous size distribution (Figure 8 (a) and Table 2).”
Order of the figures have been changed as requested.
Line 321 : the surfactant was removed by refluxing in HCl/EtOH or NH4NO3/methanol. This step should be described properly in the experimental part of the paper. Furthermore, the authors did not explain why they tried two strategies to remove the surfactant and whether one strategy is better than the other.
Both methodologies work perfectly well. However, after some experimentation and only because of the simplicity of the method, we recommend the mixture NH4NO3 to remove the surfactant.
The following sentence has been added:
L 349: “(treatment with HCl/EtOH similar results)”
Suggestion to the authors : a table with the ssa and pore volume would be more helpful for the clarity of the manuscript than the curves in Figure 8.
We thank the reviewer for such a suggestion. Table 2 contains the values of the parameters extracted from the graphics shown in figure 8.
Line 369 : three and not there
Changed as requested.
Questions to the authors :
Paragraph 2.16 : the release experiments should be described in detail. For example, the author should explain how they have withdrawn Ataluren without withdrawing MSN loaded with Ataluren?
We agree with the referee and the following paragraph has been added to the text:
L 211: “10 mg of MSNs loaded with Ataluren were placed inside an Eppendorf. Then, 1.5 mL of citric acid 0.1M / sodium phosphate (Na2HPO4·12H2O) 0.2 M buffer (pH=7.4) was poured into the latter. The Eppendorf was sonicated until a clear dispersion was obtained. After that, the solution was allowed to stand in an incubator for an specified time. The resulting precipitated was isolated by centrifugation at 13000 rpm for 13 min. 1.3 mL of the supernatant was extracted and stored in order to calculate the Ataluren concentration through UV-vis spectroscopy at 244 nm. The Eppendorf was refilled again with 1.3 mL of the previous buffer. All this operation was performed 5 more times (24 h of release).”
Line 229 : what « good chemical stability » means ? What kind of test did you use to check the stability ?
In this context, chemical stability means that the functionalization remains essentially unaltered, in terms of reactivity, when the NP’s were stored in the fridge for weeks. In other words, the reactivity of the NP’s with FITC was essentially the same with fresh samples than with the new ones.
The following sentence has been added:
L 250: The resulting MSNs show good chemical stability and can be stored at room temperature indefinitely (checked by reaction of MSN-(NCS) with FITC).
Line 238 : how can you explain such a size difference between MSN-NH2 and MSN-NCS. The PDIs are also strongly different suggesting agglomeration due to the surface modification. Thank you for your comments.
Certainly, we agree with the reviewer that this difference in size and PDI can be, at least to some extent, due to aggregation. However, differences on the hydrodynamic radii are usually observed when the superficial functionalization of the NP is changed. The result of the DLS measurements strongly depends on the way how the solvent interacts with the surface of the nanoparticle. Thus, the functionality onto the nanoparticles will define this interaction. In any case, TEM images display regular, homogeneous and round shaped. Examples on how the hydrodynamic radius depens on the nature of the surface funtionalization can be found in the following references: J. Mater. Chem., 13476–13482, 2011 and Langmuir, 12909–12915, 2012. (cf. the comment regarding differences between DLS and TEM mesurements.)
Line 239 : Could you comment on the negatively charged MSN-NH2 ? We could expect either neutral charged or positively charged nanoparticles due to the presence of amine groups on the surface of MSN.
In this work, aminated MSN have been synthetized by co-condensation of TEOS and APTES following a reported methodology (cf. J. Mater. Chem. 2009, 1193-1340, 2009). The zeta potential expected for (MSN-OH) is about -30 mv. A range of potential from-10 to 0mV is described for the aminated MSN. (Chem. Mater., 7207–7214. 2008). The value of the zeta potential depends on the balance among amino groups onto the surface of the nanoparticles and the silanol groups, and the solvent. The values obtained in this work are in agreement with those reported in the literature (reference 44).
Figure 3 : it seems that the main peaks is slightly shifted to larger values of 2 theta after the grafting. Can you comment on that ? Besides in the table S1, we can see that the DLS size is much higher that the TEM size especially when the particles are 50 nm in diameter. Can you explain such a difference ?
Figure 3 shows SXRD of MSN-(NCS) and MSN-(NH2)i-(NCS)o. Both nanoparticles have been functionalized, but the later contains both amino groups and SCN groups. According to the literature, the intensity and position of the signals of SXDR depends on the functionalization within the pores of the nanoparticle. Therefore, these modifications on the intensity and position of the peaks are consistent with the differential functionalization of the nanoparticles. Examples of such phenomena are reported in: Phys. Chem. Chem. Phys., 13882, 2015 (p.13884), New. J. Chem. 6017, 2014 (p. 6019, Fig 1), Nanoscale Res. Lett., 1–19, 2012, RSC Adv., 9546–9555, 2015, J. Phys. Chem. C, 18358–18366, 2012 (p. 18361, Fig 3), Langmuir, 2986–2996, 2012 (p. 2988, Fig 1).
DLS measures the hydrodynamic radius, which depends on the conditions of the measurement (solvent, pH, concentration,…), whereas TEM values are obtained as the median value of the observed size of the NP's. Thus, these discrepancies are due to the methodology used to determine the radius. Please, see the following references: Nanoscale Res. Lett., 12:74, 2017, Langmuir 8984, 2019 (p. 8988 Table 1).
Additionally:https://www.researchgate.net/post/Why_does_the_particle_size_differ_in_TEM_and_DLS_measurements.
Line 253 : The authors should explain the decrease of the sorption propoerties of MSNs after grafting the isothiocyanate moiety : decrease of ssa and pore volume, while the pore size remains constant.
The isotherms depends on the functionalization of the surfaces. These differences are ascribed to the presence of functional groups onto the walls of the nanoparticles (in terms of polarity, bulkiness,…). However, according to TEM images and the presence of type IV isotherms the regular mesoporous structure is retained. Please, see for instance: Phys. Chem. Chem. Phys., 13882, 2015 (p.13884), Langmuir 8984, 2019 (p. 8988 Table 1), Langmuir, 2986–2996, 2012 and J. Mater. Chem., 13476–13482, 2011. Regarding the pore diameter, the referee is correct. The value is slightly smaller in MSN-(SCN) than the one recorded for MSN-(NH2). However, the difference falls within the error of mesurement. Interestingly, the difference is more evident in the case of 50 nm MSN’s than in the bigger ones.
The following sentence has been added:
L 276: “These decreases in both the surface area and the pore volume are consistent with the functionalization of the pores of the nanoparticle.”
Line 263 : The authors should better describe the IR spectra and especially the double band 2100 cm-1. Besides, the different IR curves do not correspond to MSN-NH2 or MSN-NCS. Why ?
The correct introduction of isothiocyanates moieties can be inferred by the presence of a band centered around 2100 cm-1, which is ascribed to the NCS group.
The following sentence has been added:
L 287: “The successful functionalization of the MSNs is supported by the presence of two characteristic absorption bands around 2100 cm-1 in the FT-IR spectrum ascribed to the isothiocyanate group (Figure 5).”
The referee is correct. Caption changed as requested:
“Figure 5. FTIR spectra of MSN-(NH2) (CTAB), MSN-NH2, MSN-(NH2)i-(NCS)o(CTAB) and MSN-(NH2)i(NCS)o.”
Line 346 : What the author mean by “indefinitely”. How long time did the author let the MSNs at room temperature before checking that the porous properties of the materials did not change? Besides, what do you mean by room temperature? In ambient air?
The following sentence has been added:
L 373: “about 3 months at room temperature (23 ºC).”
Figure 11 : the authors have to explain the term %release Ataluren. How did they measure the total amount of drug in the particles ?
The following equation has been added to the ESI:
The loading of the particles were determined by difference of the total amount of the drug and the drug solubilized in the washings.
The following sentences have been added:
L 218 Ataluren release has been calculated using equation S1 (cf. SI).
L 207: “The washings were collected and the concentration of Ataluren was determined by UV-vis spectroscopy at 244 nm to calculate the loading of the drug by difference.”
The author claim that the drug is released faster in the presence of a shorter PEG chain. Did the author try to determine the amount of PEG grafted with the different precursors? The information would be of great importance to better conclude on the influence of the chain length on the drug release even if we can easily expect that short chains will close the pores less easily that large chains.
Can the author comment on the slower drug release in the presence of molecule 10 than with molecule 9 whereas molecule 09 is bigger?
Certainly. According to previous work (Drug Delivery, 1137, 2018), the amount of amine groups onto the surface of MSN-NH2 is estimated to be around 2·10-7 mol NH2/mg MSN, which eventually corresponds to the same amount of NCS, once this functionality is introduced to the MSN. Then, a slight excess of each amine (8, 9. 10) is added to MSN-NCS in order to achieve a complete functionalization. The reaction between amines and R-NCS is well-known to be complete. Therefore, it is assumed that about 0.002 mmol of 8, 9, 10 /mg MSN are introduced onto the surface of the nanoparticle.
Regarding the influence of the chain length, in the case of 10 ((2-aminoethyl)trimethylammonium) we hypothesized that the “gradual” release of the drug is due to the interaction between the carboxylic acid functionality and the ammonium salt, which slows down the burst of the drug. In fact, the concept of self-gated MSN have been reported in the literature (Nanoscale, 17063, 2017, J Colloid Interface Sci. 345, 2018). In contrast, in the case of PEG’s, the chain length is the relevant factor to explain the effective blocking of the drug.
Reviewer 2 Report
The authors have provided a method to synthesize isothiocyanate functionalized MSNs for applications in drug delivery. The work shown in this manuscript is well described and mostly presented well. Comments are as follows:
(1) Some figures are of low quality and hard to read. Please improve.
(2) Be consistent with graphs used within the manuscript (figure 5 and 6 for example).
(3) Does figure 11 have error bars?
Author Response
Referee comments nb 2:
The authors have provided a method to synthesize isothiocyanate functionalized MSNs for applications in drug delivery. The work shown in this manuscript is well described and mostly presented well. Comments are as follows:
(1) Some figures are of low quality and hard to read. Please improve.
(2) Be consistent with graphs used within the manuscript (figure 5 and 6 for example).
We agree with the reviewer. All figures have been updated.
(3) Does figure 11 have error bars?
Errors bars have been introduced as requested.